# Implementing Dual Base Stations within an IoT Network for Sustaining the Fault Tolerance of an IoT Network through an Efficient Path Finding Algorithm

**DOI:** 10.3390/s23084032

**Published:** 2023-04-17

**Authors:** J. K. R. Sastry, Bhupati Ch, Raja Rao Budaraju

**Affiliations:** 1Department of ECM, K L Deemed to be University, Vaddeswaram 522302, India; bhupati@kluniversity.in; 2Oracle America Inc., Scottfield Street, Dublin, CA 94568, USA; rajaraob@yahoo.com

**Keywords:** fault tolerance, IoT networks, base stations, remote communication

## Abstract

The IoT networks for implementing mission-critical applications need a layer to effect remote communication between the cluster heads and the microcontrollers. Remote communication is affected through base stations using cellular technologies. Using a single base station in this layer is risky as the fault tolerance level of the network will be zero when the base stations break down. Generally, the cluster heads are within the base station spectrum, making seamless integration possible. Implementing a dual base station to cater for a breakdown of the first base station creates huge remoteness as the cluster heads are not within the spectrum of the second base station. Furthermore, using the remote base station involves huge latency affecting the performance of the IoT network. In this paper, a relay-based network is presented with intelligence to fetch the shortest path for communicating to reduce latency and sustain the fault tolerance capability of the IoT network. The results demonstrate that the technique improved the fault tolerance of the IoT network by 14.23%.

## 1. Introduction

Fault tolerance of any network is generally defined as the reliability of the availability of the IoT system in working conditions. The more reliable the IoT network, the more tolerable the network, leading to the high acceptance of such a network. The IoT networks relating to mission criticality systems must be tolerable to the extent of 99% [1].

IoT networks use networking topologies (butterfly, Crossbar, Hybrid) in different layers connecting small devices (sensors, actuators, controllers) and big devices such as high-end servers, base stations, gateways, and splitters. Small devices often fail and induce different faults, rendering the entire IoT network less fault tolerant [2].

The devices within an IoT system are heterogenous and are driven through different protocols requiring conversions, speed matching and the use of several sophisticated algorithms for dealing with data transmission for the purpose of performance enhancements [3,4,5,6,7].

The fault tolerance of an IoT network can be measured in terms of success rate, failure rate, false alarm rate, and power depletion rate [8] and the computation of these metrics. In addition, different computation models are to be used, which include FTA (Fault Tree Analysis) [9] for linear models, probability models [10], hybrid models [11] that combine linear and probability models, empirical model [12], and bipartite flow graph modeling [13].

Small devices tend to fail, inducing different faults, sometimes rendering the entire IoT system non-operational and producing unpredictable behaviour. The faults that are generally induced in the IoT system include Cascading faults [14,15], Pattern faults [16], and device-to-device communication faults [17]. Various faults occurring within the devices have been reviewed by Norris et al. [18,19].

Several methods have been implemented to enhance the fault tolerance of an IoT system in the presence of the faults mentioned above. The methods aim to enhance the robustness [14], carry complex event processing [20], minimization of bandwidth [21], alternate Networking butterfly [22], Crossbar [23], and handle demand uncertainty [24]. S. Kumar, P. Ranjan et al. [25] presented an artificial-intelligence-based method for analyzing the fault tolerance level of the IoT network in the presence of network failures within software-defined IoT networks.

The cluster heads transmit the sensed data routed by the devices through base stations to the microcontrollers due to the existence of both devices over long distances. Cellular technologies are used to effect communication between devices. A base station is placed to provide a nearby spectrum for the cluster heads to communicate with the microcontrollers to transmit data and for receiving the actuating commands. The network becomes isolated when the base station fails, and the fault tolerance of such a network is zero. It will be dangerous when the fault tolerance of a mission-critical system becomes zero.

K. M. Malarski et al. [17] presented a D2D-enabled Fault-tolerance in Cellular IoT. They have yet to consider the issue of the remoteness of the second base station and also based their discussion on a single relay system which generally cannot support a huge number of cluster heads.

Building an IoT network considering more than two base stations, is very complex. When a base station is placed in a distant location, the latency is bound to increase, affecting the performance of the IoT system. The placement and connectivity of cluster heads and the microcontroller with the base station is a complex issue that must be addressed to minimize the latency. There should be alternate paths of communication to cater for the failure of one of the base stations.

It is necessary to build alternative methods of communication between the cluster heads and microcontrollers through a pair of redundant base stations, which are situated between cluster heads and the microcontrollers. A networking topology and routing algorithm must be developed to connect the cluster heads to the microcontrollers through a remotely situated base station. The communication between the base stations and the microcontrollers is proposed to be undertaken in peer-to-peer mode using four different channels to synchronize the communication speeds between slow-speed microcontrollers and high-speed base stations.

The latest 5th-generation networks have provided significant growth in low latency communication and high speeds, which support many missions’ critical applications [26,27], such as industrial automation, e-health, smart cities and autonomous cars, imposing stringent network performance requirements from communication networks.

Healthcare-related mission-critical applications must have high availability, dependability, and low end-to-end latency [28]. For machine-type communication, the Internet of Things uses two cellular communication technologies: narrowband-IoT (NB-IoT) and LTE-M, which were created for low-power consumption devices and include expanded coverage, deep door signal penetration, and high availability [29,30,31]. These channels must adhere to duty cycle limitations, use permitted spectrum, and minimise interference from nearby devices. These conditions can be met using the Device-to-Device (D2D) paradigm to prevent complete disconnection between Cellular IoT (CIoT) devices [32]. Although NB-IoT does not entirely support LTE, it only uses 180 kHz of available bandwidth and offers three deployment options: in-band, guard band, and standalone.

When two CIoT devices are close to one another, and one is still connected to an evolved node-b (eNB), it is possible to establish a direct communication link using D2D and relay packets from the failed device to the network infrastructure [33]. D2D might also be used to offload traffic from the CIoT network. Reducing energy consumption and E2E delay can be accomplished by offloading device traffic from the network [34,35].

The use of a separate D2D network will help in improving the fault tolerance of the network. However, the real challenge is that the base stations are remotely situated from the devices making it more complicated to establish communication between the cluster heads and the remote controllers through base stations. More efficient networking that provides alternate paths of communication is required. The shortest path must be selected for communication, so that the data move to the service server with the least latency.

There is a need to establish communication between the slow-speed and high-speed devices and networks, which leads to heavy failures. Heterogeneity results in significant failures and thus needs to be managed. Communication between the cluster heads and remotely situated microcontrollers is affected through base stations. The cluster heads and the microcontrollers communicate with the base stations through cellular communication. The risk of failure of the IoT network is very high due to failure of the base station and the presence of heterogeneity in the communication protocol. The remote base stations are out of sight to the devices that sense and transport data.

A specialized networking topology is required to connect the devices to a redundant base station remotely situated from the cluster heads. Redundant base stations enhance the fault tolerance of the IoT network. The base stations communicate with the microcontrollers in peer-to-peer mode using parallel communication so that the speeds match. There should be an element of redundancy in the network to accommodate the link failures. Furthermore, communication must be done via the shortest path with the least traffic so as to minimise latency.


**The following research questions are answered in this paper:**
How will the longevity of the IoT network be affected due to the use of a single base station in the IoT network?How to determine fault-free and fast responsive paths in a given IoT networkWhat parameters must be considered to decide the path to effect the fastest communication through the 2nd base station?How many base stations are required to guarantee a 100% fault tolerant system



**Major Contributions of this research**


A method to determine the shortest and fastest fault-free path for data transmission from cluster heads to microcontrollers en route to the redundant base station.A method to facilitate communication between the base stations and the microcontrollers.A method to convert a networking diagram to a Fault tree Analysis diagram considering different networking topologies used to connect both the device layer and the controller layer.


**Outline of the paper**


The rest of this paper is presented in Section 3, Section 4, Section 5, Section 6, Section 7, Section 8, Section 9 and Section 10. The related work and the GAP are presented in Section 2. In Section 3, the overall method used for improving the fault tolerance of the IoT network through modifications effected in the controller layer is presented. Section 4 presents an updated and improved IoT network up to the device level, along with its fault tolerance diagram and computations. The revised IoT network and the changes effected in the controller layer of the IoT network are explained in Section 5. Section 6 explains the networking topology and the devices used in the network connecting the cluster heads and the base station. A novel fault-free, shortest and fastest path-finding algorithm is discussed in Section 7. Results of the experiments on the revised network and a discussion on the same are presented in Section 8 and Section 9, respectively. Conclusions and future scope are presented in Section 10.

## 2. Related Work

D. Koziol et al. [36] proposed that achieving QoS between the base station and the remote user equipment (UE) requires resolving many trade-offs regarding signalling overhead, implementation complexity and overall delay. They did not recommend any alternate mechanisms to ensure fault tolerance is maintained in the event of failures while achieving the required level of QoS.

Skorin-Kapov [37] presented Machine-Type Communication (MTC) which considers simplification of the channels and interfaces such that the communication system is suitable for NB-IoT and LTE-M-based applications. They did not consider any specific networking topology or failure situations.

C. Min et al. [38] expressed that MTC created an interest in D2D applications, especially in the automotive sector. They did not consider any specific networking topology or failure situations.

According to Hunukumbure et al. [39], adding D2D to cellular infrastructure will increase reliability, reduce power usage, and prevent network congestion. They suggested a random-access process that uses broadcast messages to announce the D2D mode of communication so that UEs can directly talk with one another, utilising CSMA with collision avoidance. Yet, the endeavour must consider IoT devices and the various resource limits they impose. The concerns with the distance between the base stations and the equipment on either side of the stations must still be considered.

A D2D approach for improving battery life and the accessibility of Cellular IoT (CIoT) deployment was proposed by J. Lianghai et al. [40]. With the help of the gathered environmental data, the network oversees and maintains the assignment of the UEs to the devices (remote devices, relays) (e.g., battery level and position). The authors create new signalling behaviour allowing UE attachment, transmission mode (re)configuration, and uplink data delivery. Applying a D2D solution controlled by a core network could address RAN (Radio Access Network) failure issues more effectively.

Al-Salihi NK [41] proposed an Internet of Things (IoT)-based position-fixing system instead of GPS as they are found to be useful in tracking the daily activities of children, the elderly and vehicle tracking. They proposed a redundancy-based model for improving the fault tolerance of the IOT-based position-fixing system. However, they did not account for alternate communication paths in the event of a failure occurring.

Bhupathi et al. [42] presented different metrics for computing the fault tolerance of the IoT network. They also presented [11] a crossbar network topology to connect the clusters to the cluster heads. Moreover, they implemented a method to predict the fault that a device will inject due to the power depletion rate. The devices are isolated before the fault can be injected into the system.

Many have proposed algorithms to find the shortest path from a source node to the sink node.

Daniel Foead et al. [43] presented a complete review of different variants of A*-based search algorithms and opined that the algorithm fails as the sizes of the network increase and dynamically change.

F. Xia et al. [44] presented a review of different variants of random walk algorithms, which are meant to find the shortest path in the network. They proposed that one has to select a variant of the algorithm string for a specific application. The algorithm, as such, does not consider the existence of the loops in the network.

Niranjane P. B. et al. [45] presented a comparison of variants of Yen’s algorithm for finding k-simple shortest paths, which is based on the number of deviations that the network contains. However, the need for k-shortest paths rarely arises when pruning the paths is achieved to eliminate the failure paths from the system.

Kyle E. et al. [46] studied the choice of a graph search algorithm to find the shortest path in a directed relation graph with error propagation (DRGEP and have compared the method with other algorithms that include depth-first search, basic and R-value-based breadth-first search (RBFS), and Dijkstra’s algorithm and found that Dijkstra’s algorithm combined with coefficient scaling approach most accurate results when applied to bio application.

Kalyan Mohanta B. P. et al. [47] presented a comprehensive review of the existing k-shortest algorithms and showed the computational efficiency of each of the algorithms. Andrej Brodnik et al. [48] presented an all-pairs shortest path algorithm for directed acrylic graphs and arbitrary edge lengths. Muteb Alshmmari et al. [49] proposed an algorithm (single source shortest path) for dynamic graphs with large change frequency.

The methods proposed in the literature did not focus on the issue of failure of the main base station and the need for a second redundant base station, and the way the second base station is to be connected to cluster heads considering distance limitations and non-crossover of the communication spectrum. None have attempted to determine the fail free shortest and fastest path that should be used for affecting communication between cluster heads and the second base station.

## 3. Methodology

A flow diagram depicting the execution of the proposed methods is shown in Figure 1. The blocks relating to metrics development, initial prototype development and its FTA development, implementing a crossbar network in the device layer, implementing fault detection and isolation method in the sensing and actuating devices to counter the possible fault in injection in the device layer and its related FTA model and the improvement in fault tolerance capability of the IoT network were explained by Bhupathi et al. [11,42]. Further to the implementations carried out in the device layer, a second base station has been added to the controller layer. The second base station has been connected to the cluster heads through an intelligent-relay-based network with built-in redundancy to tackle the issue of remoteness of the base station from the cluster heads. An intelligent pathfinding algorithm is implemented in each relay to find the shortest path with the least traffic to significantly reduce the latency. An FTA diagram for the newly introduced network is developed and combined with the FTA diagram of the prototype model. Fault tolerance values have been computed through the generation of a fault table. A comparison of the stage improvements in the fault tolerance of the IoT network due to the implementation of different methods in the device layer and the controller layer has been presented.

## 4. The Updated IoT Network

An IoT network with changes at the device level is presented in Figure 2. The updated IoT system implements the following in the device layers to enhance the fault tolerance of the network [11,42].

Implement a fault detection system in the device layer that detects possible power-related faults and then isolates the faulty devices.Establishing a crossbar network between the cluster heads and the device clusters to provide alternative redundant communication paths.Develop redundant networks using different topologies connecting base stations and cluster heads.Connecting the base stations to the microcontrollers in peer-to-peer mode.A method to compute fault tolerance considering linear and probability models.Connecting several controllers to a single services serverConnecting a services server to a gateway en route to the internet connecting the cloud.

The devices in a cluster are linearized and connected to a cluster head free from sensing or actuating function. A crossbar network connects the linearized clusters’ outputs to the cluster heads. The cluster heads are connected to a base station in a parallel computing mode, and the base stations communicate with multiple microcontrollers in a peer–peer computing mode.

The microcontrollers are connected to the server of the service in many-to-one mode. The services server receives the requests from the devices or the users, executes the service-related code, and transmits the results back to controllers or the user.

The revised IoT network with changes made in the device layer is shown in the Figure 2.

For developing a fault tree of the sample IoT network, the crossbar network is replaced by a single device whose success rate is computed using probability models connected with the crossbar network [11]. The success rate of such a network is computed as 0.842. The fault tree of the residual network and a table showing the success rate computations are generated through the algorithms presented by Bhupathi et al. [11]. The fault tree generated for the updated network is shown in Figure 3.

From the sample network, the single base station is the real bottleneck and forms the most vulnerable areas of failure of the entire IoT network. Any fault accruing in this patch will disconnect the entire network, and the fault tolerance of such a network becomes zero. The need for implementation of redundancy of the base stations thus arises.

## 5. Revised IoT Network

The sample IoT network has been modified, as shown in Figure 4.

The following changes have been made in the controller layer.

A second base station, which is remotely situated due to spectrum reasons, has been added. The second base station is connected to cluster heads using a separate network established by relays/switches placed strategically because of the distant locations. Two layers of relays have been considered keeping in view the maximum connectivity distance to be 100 km.An algorithm is implemented that finds the shortest distance from a source node to a sink node and ensures that the traffic is minimum in that path. The number of bytes to be transmitted over a path is considered as the decision to select the path.Parallel communication is implemented to establish communication between the first base station and the cluster heads and between the base station.A relay-driven network with built-in redundancy establishes communication between the second base station and cluster heads. Parallel communication affects the communication between the 2nd base station and the controller.

This paper discusses ways of improving the fault tolerance of the communication between the cluster heads, base stations and microcontrollers. This paper focuses on introducing topologies that connect the cluster heads to the base stations so that fail-free operations are carried out within the least possible response time, which is related to traffic/communication distance. Here, a path with minimum traffic/distance is chosen for communication.

## 6. The Network between the Cluster Heads and the 2nd Base Station

The second base station is situated very far from the cluster heads due to the requirement of avoiding spectrum collisions. A separate network with built-in redundancy is required to cater for the failures affecting communication between cluster heads and the base station. Figure 5 shows the network between the cluster heads and the second base station.

Nodes 1, 2, 3, 4 are cluster heads. Nodes 5, 6, 7 are first layer relays, and nodes 8 and 9 are the intelligent relays. Node 10 is the base station. Redundancy is maintained between the cluster heads and the first layer’s relays and between the first and second layers, as is the case with second layer relays and the base stations. Redundancy among the nodes is achieved by making available two paths from one node to the next superseding nodes. Each node is intelligent such that the node can run a path-finding algorithm.

The FTA equivalent of such a network can be developed using the linear cluster concept. No probability model can be developed as the network follows no specific structure. An algebraic path-finding algorithm is implemented in each node to select a path based on the distance and traffic on a specific path.

## 7. Pathfinding through the Algebraic Method

Step-1 Capture the network

The following algorithm finds all the paths between the source and sink nodes and then finds the path that requires transmitting minimum data per KM distance. The steps involved are described below:

The algebraic method primarily involves capturing the network in terms of precedence matric, which predominantly represents the network’s structure. For the network shown in Figure 5, the precedence relationships are shown in Table 1. The table shows the number of bytes of data to be transmitted from a node at a point in time, the transmission speeds used by each node, the amount of time it takes to transmit the data, the preceding node and the distance of the preceding node from the current node. The details of the preceding connected nodes are captured for every existing node in the network.

Step-2 Capture traffic at the nodes

Based on the number of bytes transmitted from the base nodes and the number of bytes received at the base station, estimate the number of bytes yet to be transmitted at each node in the network. Every node is a relay/intelligent communicating device; all relays are assumed to communicate at the same speed of 11 Mbps. The size of the data pending transmission is recorded and the data distribution is performed based on equal proportions considering the number of outgoing paths from a specific node. The distance between any pair of nodes is known and recorded. Table 1 shows that the node with no preceding nodes is a starting node, and the node not succeeding is the sink node or terminal node.

Step-3 Generate algebraic equations using the precedence table

Considering the precedence of the nodes, the following algebraic equations are generated.

Initial nodes = 1, 2, 3, 4

Sink Node = 10
(1)1→5+6
(2)2→5+6
(3)3→6+7
(4)4→6+7
(5)5→8+9
(6)6→8+9
(7)7→8+9
(8)8→10
(9)9→10

Step-4 Generate algebraic equations using the precedence table

For each source, the node implements substitutions and generates network paths by including node-specific equations. For example, include Equations (5) and (6) in Equation (Equation 1) Include Equations (5) and (6) in (1) 1→ 5 (8 + 9) + 6 (8 + 9)
(10)1→58+59+68+69

Include Equations (8) and (9) in Equation (Equation 10).
(11)1→5810+5910+6810+6910

Step-5 Generate paths from the source nodes

The paths generated from each source node are shown in Table 2. One can see that 16 Paths have been generated.

Step-6 Path pruning

If any node in the network fails, say node 6, the path containing node 6 is ignored and marked as pruned. The pruned status of the paths due to the failure of node 6 is shown in Table 2. A total of 8 Paths now remain for communication.

Step-7 computes the distance and extent of data to be transmitted for each source node

For all the non-pruned paths, the total distance and the extent of data transmitted are computed as shown in Table 3. It is seen from Table 3 that amount of data to be transmitted is the same due to the principle of distributing the data equally among all outgoing paths. The size of the data to be transmitted per km distance is shown in the table based on which the path that should be selected is decided.

Step-8 Path Selection

The path to be selected is based on the source node, and the number of bytes to be transmitted per km is minimum. In the case of source node 1, path-1 (1 + 5 + 8 + 10) is selected for transmission as it has the lowest number of bytes to be transmitted. Similarly, suitable paths for other source nodes are selected.

## 8. Results

Developing fault tree for the revised IoT network.

The crossbar network between the controllers and services servers is replaced by a single device assigned with a success rate that is the same as the crossbar network’s success rate using its related probability model. The additional network added to the network is converted to an FT diagram using AND/OR conditions based on the precedence and data flow designed through building redundancy in the network. The modified FT diagram related to the revised IoT diagram is shown in Figure 6.

The FTA diagram is generated using the algorithm presented by Bhupathi et al. [11]. Similarly, the crossbar network between the devices and the cluster heads is replaced by a single device assigned with a success rate the same as the crossbar network’s success rate using its related probability model. These transformations convert the FTA diagram into a linear model.

### Success Rate Computations

Bhupathi et al. [11] presented the use of an algorithm to generate the success table, given the FTA diagram as the input. The generated success table is presented in Table 4. The success rates of every device are computed using its precedence relationships with other devices. From the table, it can be observed that the success rate of the revised IoT network is 0.980.

## 9. Discussion

The algebraic algorithm built into every node in the additional network to connect the second base station to the cluster heads performed well compared to its nearest pathfinding algorithm (single source shortest path). A comparison of the algorithms is shown in Table 5.

Several pathfinding algorithms proposed in the literature have been surveyed and the same is compared with the algebraic pathfinding algorithm presented in this paper. The comparison considers the number of nodes, edges, shortest paths, path pairs and several elements. The comparison is carried out considering the network shown in the revised IoT diagram. From the table, it can be observed that algebraic methods require fewer operations for selecting a path for data transmission, given that a source node from the transmission is initiated.

The failure rate of a base station is negligible. The success rate is around 98.0%. The entire IoT network will malfunction when a base station fails. The probability of which is 0.98. To ensure a failure-free situation (100% success rate), adding a second base station is necessary, which requires a different type of networking topology because the distance between the cluster head and the second base station becomes a major issue.Adding one redundant base station is sufficient as it provides a 100% success rate. With the addition of more base stations, the cluster heads will become overloaded, which leads to a depletion of response time due to increased latency.

The success rate of the revised IoT network is 0.948 when compared with an IoT network that caters for changes in the device, the fault rate of which is fixed at 0.827. A comparative analysis of improvements in the fault tolerance in the IoT network achieved with changes made into different layers of the IoT network is shown in Table 6. With the changes made to the IoT network, the fault tolerance level of the IoT network is increased by 14.63%.


**Alternative Justification**


The fault tolerance capacity of both the sample network and the revised network are computed considering different failure conditions, including communication failure between a cluster head and the base station. The computations are shown in Table 7. The table shows that the revised network retains the FTA level even though some failures happen in the communication paths that connect the cluster heads to the base stations. The combined failure of the IoT network, considering the failures between cluster heads and the base stations and the failures between the base stations and the controllers, improved from 0.45 to 0.64, a 42% improvement.

## 10. Conclusions and Future Work

### 10.1. Conclusions

The fault-tolerance capability of an IoT network is critical, especially when mission-critical systems are built using IoT technologies.The fault-tolerance capability of an IoT network can be enhanced by making suitable changes in each of the layers of the IoT. This paper focuses on enhancing the fault-tolerance capability considering the controller layer.A single base station-based IoT is risky and unsuitable for implementing mission-critical systems.The fault-tolerance capability of the IoT network improves when a redundant base station is added, and the same is connected via an intelligent-relay-based network. The success rate of the revised IoT network increased from 0.827 to 0.948, which is a 14.23% improvement.Considering both parts of the network, the combined success rate, which includes the path from cluster heads to the base station and the path from the base station to the controller, improved from 0.45 to 0.64, a 42% improvement. The latency of communication using such a network is minimum.The path-finding algorithm implemented in the intelligent relays requires fewer operations than any other algorithm presented in the literature.

### 10.2. Future Work

Further enhancement in the fault tolerance of the IoT network can be carried out at the controller level, services layer, and gateway layer by incorporating suitable changes considering the devices and the connectivity between devices in those layers.

## Figures and Tables

**Figure 1 sensors-23-04032-f001:**
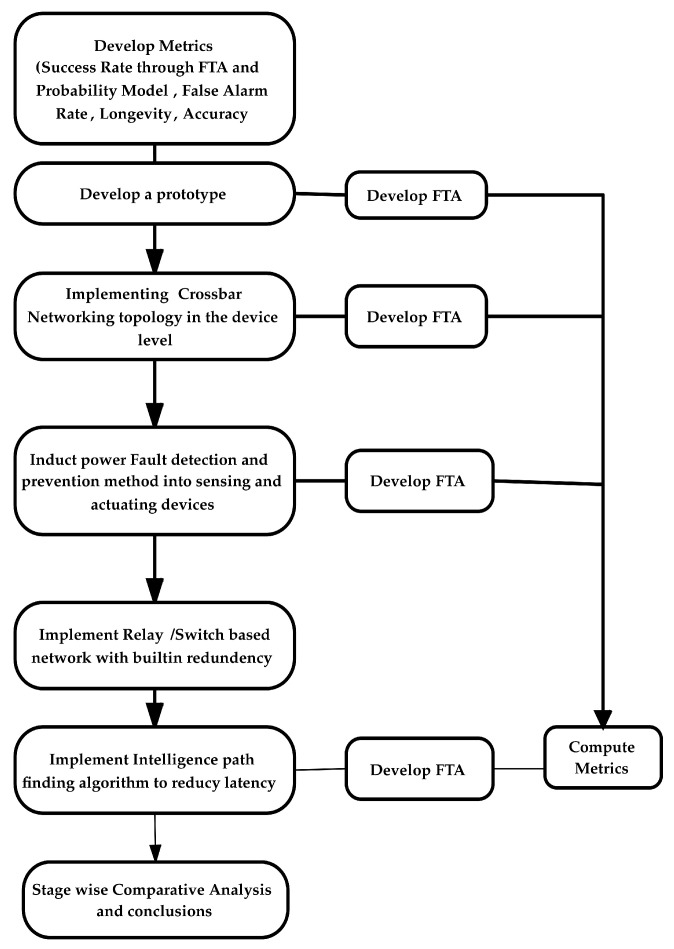
Overall method for improving the fault tolerance of the IoT network.

**Figure 2 sensors-23-04032-f002:**
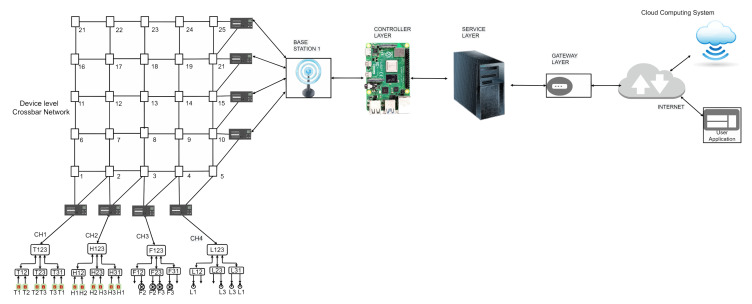
Sample IoT network with a single base station.

**Figure 3 sensors-23-04032-f003:**
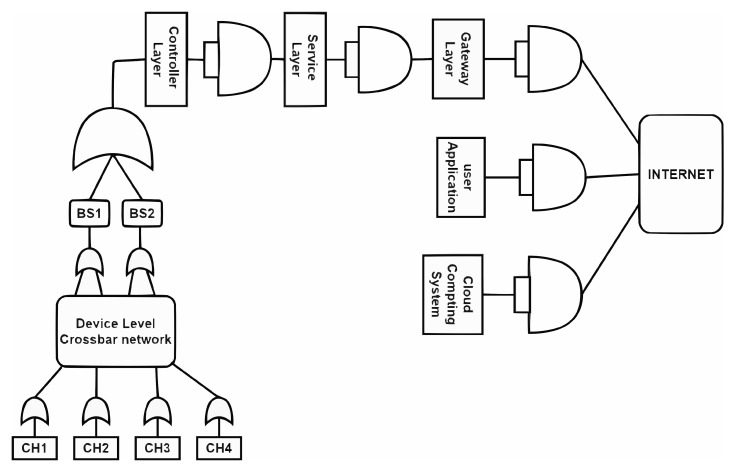
Fault tolerance diagram for the sample IoT network.

**Figure 4 sensors-23-04032-f004:**
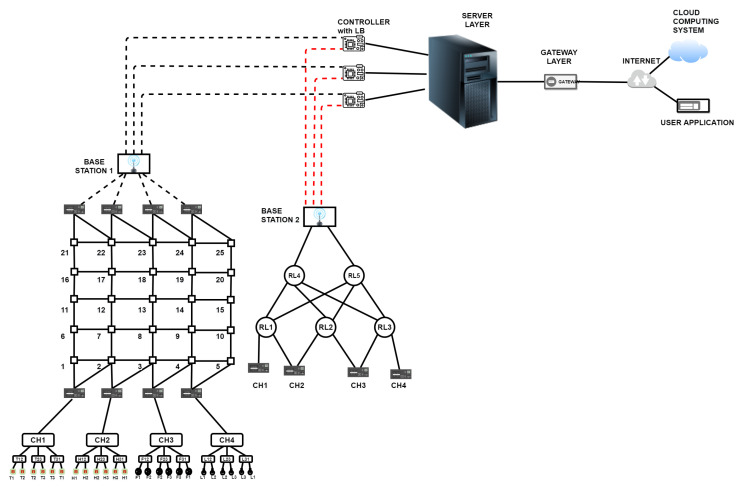
Revised IoT diagram with redundancy created at the base station and controller levels.

**Figure 5 sensors-23-04032-f005:**
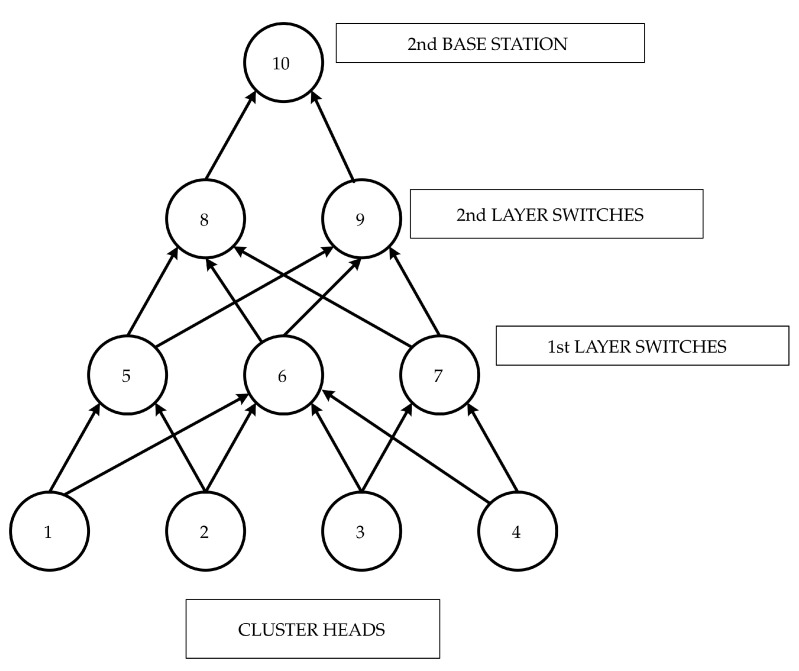
Networking Topology connecting cluster heads and redundant base station.

**Figure 6 sensors-23-04032-f006:**
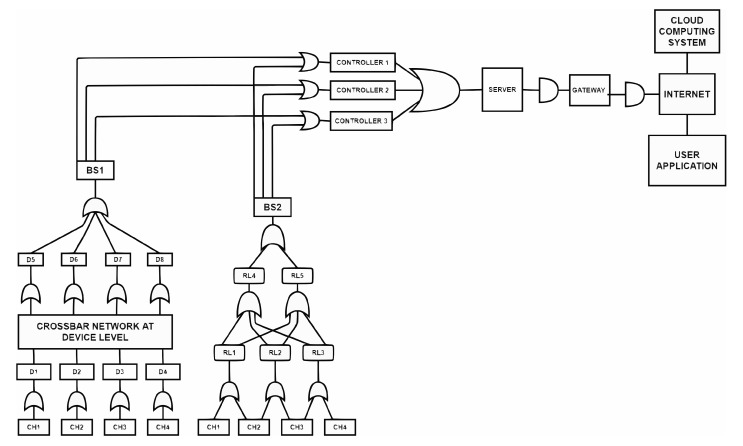
FTA model for revised IoT network.

**Table 1 sensors-23-04032-t001:** Precedence Matrix for the Example Network Diagram.

Node	Type of Node	Number of Bytes to Be Transmitted	Transmission Speed on Mbps	Latency in ms	Preceding Node	Distance in km
1	Cluster Head-1	3,000,000	11	0.273	0	0
2	Cluster Head-2	3,000,000	11	0.273	0	0
3	Cluster Head-3	3,000,000	11	0.273	0	0
4	Cluster Head-4	3,000,000	11	0.273	0	0
5	SWITCH-5	1,500,000	11	0.136	1	3
5	SWITCH-5	1,500,000	11	0.136	2	2
6	SWITCH-6	1,500,000	11	0.136	3	1
6	SWITCH-6	1,500,000	11	0.136	2	2
6	SWITCH-6	1,500,000	11	0.136	3	2
6	SWITCH-6	1,500,000	11	0.136	4	1
7	SWITCH-7	1,500,000	11	0.136	3	4
7	SWITCH-7	1,500,000	11	0.136	4	1
8	SWITCH-8	750,000	11	0.068	5	2
8	SWITCH-8	750,000	11	0.068	6	1
8	SWITCH-8	750,000	11	0.068	7	1
9	SWITCH-9	750,000	11	0.068	5	4
9	SWITCH-9	750,000	11	0.068	6	3
9	SWITCH-9	750,000	11	0.068	7	2
10	SWITCH-10	2,250,000	11	0.205	8	3
10	SWITCH-10	2,250,000	11	0.205	9	2

**Table 2 sensors-23-04032-t002:** Enumerated paths commencing from all source nodes.

Source Node and Equation	Path Number	Path	Pruned Status Due to Failure of Node 6
1	Path-1	1 + 5 + 8 + 10	
1	Path-2	1 + 5 + 9 + 10	
1	Path-3	1 + 6 + 8 + 10	Pruned
1	Path-4	1 + 6 + 9 + 10	Pruned
2	Path-5	2 + 5 + 8 + 10	
2	Path-6	2 + 5 + 9 + 10	
2	Path-7	2 + 6 + 8 + 10	Pruned
2	Path-8	2 + 6 + 9 + 10	Pruned
3	Path-9	3 + 6 + 8 + 10	Pruned
3	Path-10	3 + 6 + 9 + 10	Pruned
3	Path-11	3 + 7 + 8 + 10	
3	Path-12	3 + 7 + 9 + 10	
4	Path-13	4 + 6 + 8 + 10	Pruned
4	Path-14	4 + 6 + 9 + 10	Pruned
4	Path-15	4 + 7 + 8 + 10	
4	Path-16	4 + 7 + 9 + 10	

**Table 3 sensors-23-04032-t003:** Distance and traffic computations of pruned paths.

Path	Distance in km	The Extent of Data to Be Transmitted in Bytes	The Extent of Data in Bytes to Be Transmitted/km
Path-1 = 1 + 5 + 8 + 10	2 + 2 + 3 = 7	1,500,000 + 750,000 + 2,250,000 = 4,500,000	642,857
Path-2 = 1 + 5 + 9 + 10	2 + 4 + 2 = 8	500,000 + 750,000 + 2,250,000 = 4,500,000	562,500
Path-5 = 2 + 5 + 8 + 10	2 + 2 + 3 = 7	500,000 + 750,000 + 2,250,000 = 4,500,000	642,857
Path-6 = 2 + 5 + 9 + 10	2 + 4 + 2 = 8	500,000 + 750,000 + 2,250,000 = 4,500,000	562,500
Path-11 = 3 + 7 + 8 + 10	4 + 1 + 3 = 8	500,000 + 750,000 + 2,250,000 = 4,500,000	562,500
Path-12 = 3 + 7 + 9 + 10	4 + 1 + 2 = 7	500,000 + 750,000 + 2,250,000 = 4,500,000	642,857
Path-15 = 4 + 7 + 8 + 10	1 + 1 + 5 = 7	500,000 + 750,000 + 2,250,000 = 4,500,000	642,857
Path-16 = 4 + 7 + 9 + 10	1 + 1 + 2 = 4	500,000 + 750,000 + 2,250,000 = 4,500,000	112,500

**Table 4 sensors-23-04032-t004:** Success rate computations of the revised IoT network.

Sl. No	Device	Success Rate	Gates Used For Connection	Preceding Devices				Combined Success Rate
				Device Name D1	Device Name D2	Device Name D3	Device Name D4	
				Success Rate S1	Success Rate S2	Success Rate S3	Success Rate S4	
1	Cluster Head1	0.950						0.950
2	Cluster Head2	0.950						0.950
3	Cluster Head3	0.950						0.950
4	Cluster Head4	0.950						0.950
5	D1	0.950	OR	Cluster Head1 0.950				0.950
6	D2	0.950	OR	Cluster Head2 0.950				0.950
7	D3	0.950	OR	Cluster Head3 0.950				0.950
8	D4	0.950	OR	Cluster Head4 0.950				0.950
9	Device Level CrossBar NW	0.987	OR	D1 0.950				0.987
10	Device Level CrossBar NW	Level	OR	D2 0.950				0.987
11	Device Level CrossBar NW	0.987	OR	D3 0.950				0.987
12	Device Level CrossBar NW	0.987	OR	D4 0.950				0.987
13	D5	0.950	OR	DLCB 0.987				0.987
14	D6	0.950	OR	DLCB 0.987				0.987
15	D7	0.950	OR	DLCB 0.987				0.987
16	D8	0.950	OR	DLCB 0.987				0.987
17	BS1	0.950	OR	D5 0.987	D6 0.987	D7 0.987	D8 0.987	0.987
18	RL1	0.950	OR	CH1 0.950	CH2 0.950			0.950
19	RL2	0.950	OR	CH2 0.950	CH3 0.950			0.950
20	RL3	0.950	OR	CH3 0.950	CH4 0.950			0.950
21	RL4	0.950	OR	RL1 0.950	RL2 0.950			0.950
22	RL5	0.950	OR	RL1 0.950	RL2 0.950			0.950
23	BS2	0.950	OR	RL4 0.950	RL5 0.950			0.950
24	Controller1	0.979	OR	BS1 0.987	BS2 0.950			0.987
25	Controller2	0.979	OR	BS1 0.987	BS2 0.950			0.987
26	Controller3	0.979	OR	BS1 0.987	BS2 0.950			0.987
27	SERVER	0.980	OR	Controller1 0.987	Controller2 0.987	Controller3 0.987		0.987
28	GATEWAY	0.980	AND	SERVER 0.987				0.967
29	INTERNET	0.980	AND	GATEWAY 0.967				0.948

**Table 5 sensors-23-04032-t005:** Complexity analysis of different methods used for path generation.

Serial Number	Type of Method	Complexity	Number of Operations
1	A* [43]	(ne/2)${}^{2}$	6400
2	Graph Search Algorithms [46]	(ne)${}^{2}$	128,000
3	Yen shortest paths citeref-journal45	kn + m × log m	170
4	All Pair’s shortest paths [48]	m × n + m × log ⁡n	110
5	Random walk [44]	n × e	160
6	Single Source shortest path [49]	f + f × log(f)	62
7	Algebraic method	n × Max(e) form a Node	20

where k = number of shortest paths = 16; n = number of nodes = 10; m = number of path pairs = 10; e = number of edges = 16; f = number of elements = 5.

**Table 6 sensors-23-04032-t006:** Comparative analysis of enhancements for fault tolerance in different layers of the IoT network.

Serial Number	Type of Method	Fault Tree Value
1	Prototype network [11]	0.780
2	Prototype with Changes Made in the device Levels—Proposed method [42]	0.827
3	Prototype with Changes Made in the device Level and controller Layer	0.948

**Table 7 sensors-23-04032-t007:** Analysis of Failure conditions Vs. Fault tolerance level of the network.

Parameter	Sample IoT Network	Revised Network
Number of controllers	3	3
Number of base stations	1	2
Number of paths connecting the controllers and the base stations	3	6
Number of paths connecting the cluster heads and base stations	4	16
Fault tolerance in the event of failure of connectivity of a controller with a base station	0.67	0.670
Fault tolerance in the event of failure of connectivity of a cluster head to a base station	0.67	0.948
Fault tolerance considering independent failures in the controller layer	0.45	0.640

## Data Availability

Not applicable.

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
