# Peer review of "Implementing Dual Base Stations within an IoT Network for Sustaining the Fault Tolerance of an IoT Network through an Efficient Path Finding Algorithm"

_sensors, 2023, doi:10.3390/s23084032_

Round 1
Reviewer 1 Report
Dear Authors,
What will be the effect on IoT network if we increase the number of base stations? It will be beneficial for the researchers if you can discuss this point in your paper, with some results.
Reviewer 2 Report
The authors investigated a new approach for sustaining fault tolerance is IoT networks by using the dual base station (of wireless cellular system). The second base station is connected to cluster heads of devices with a mediation of reconfigurable layered network of relays/switches keeping the maximum connectivity. They used the intelligent pathfinding algorithm is implemented in each relay to find the shortest deal with the least traffic to reduce the latency.
The work is interesting and relevant to the current topic. The reviewer recommends the publication after making minor revisions.
1) The authors presented the problem and well described the state of-art in considered area. However, in the introductory sections of the manuscript, the authors did not outline clearly enough how they intended to solve the problem and did not briefly outline the layout of the manuscript. Please add (or stress more) this information.
2) Authors make some analysis for the performances of the system in the case when some nodes ale malfunctioned, but they do not explain why such particular design of the layers in the network is effective. They do dot explained how the network was optimized. Please provide this information.
3) What is the reason to show Table 1 where for all nodes the same values are presented in all columns ? Can You just comment about these results instead repetitively tabulate the same values?
4) Quality of figure is low, in general, and their captions are in laconic.
- Please e.g. mark the cluster head (Fig.2,3), layers of switches (Fig.3).
- Correct the alignment of caption in the boxes in Fig.5.
- Please make the captions more descriptive.
5) In Eq.1-11, please consider using right-arrow symbol instead the ‘=’ sign.
6) Remove comment “* Tables may have a footer” below each table.; Authors’ affiliation on title page is doubled – please correct it.
Reviewer 3 Report
In the paper, dual base station applications have been carried out in internet of things networks.
The fault tolerance of an IoT network through an Efficient Path Finding Algorithm has been proposed.
*The main addressed problem in the paper is to improve the fault tolerance capability of an IoT network.
*IoT networks are a part of heterogeneous networks and 5G and beyond communications, so the developed study has been up to date.
*Also, the novelty in the study is to improve the fault tolerance capability of the IoT network by adding a redundant base station, and the same is connected via an intelligent relay-based network.
*The conclusions are consistent with the evidence and arguments presented and they address the main question posed.
* The references are appropriate.
Pros:
The paper has novelty and is well-designed.
The literature review is enough and discussed in the paper.
Figures and tables in the paper are related and explained in the relevant paragraph.
Finally, a relay-based network is presented with intelligence to fetch the shortest path for communicating to reduce latency and sustain the fault tolerance capability of the IoT network.
The used technique improved the fault tolerance of the IoT network by 14.23%.
Cons:
The paper has some grammar errors and typos. They should be checked carefully.
In figure 4, some writings cannot be read.
In figure 5, some writing problems have occurred.
The author may add future works.
Reviewer 4 Report
As indicated in the title, the work is an implementation of systems that have been developed before and even improvements that I consider evident. Although the work is well written, I do not identify a relevant novelty that presents benefits from the authors' proposal, using a dual base is widespread in current architectures. The conclusions are superficial and do not focus on the proposed method. In general, the document presents references to other works and it is difficult to identify the contribution of the authors, independent of the implementation and the mixture of other works.
Round 2
Reviewer 3 Report
The problems have been fixed.
The final version of the paper can be accepted.
Reviewer 4 Report
The authors explained better some elements of the proposed model, and extended some concepts, I consider that it is not a great contribution, but it is a good implementation.